# Complexity in Genetic Epilepsies: A Comprehensive Review

**DOI:** 10.3390/ijms241914606

**Published:** 2023-09-27

**Authors:** Cassandra Rastin, Laila C. Schenkel, Bekim Sadikovic

**Affiliations:** 1Molecular Genetics Laboratory, Molecular Diagnostics Division, London Health Sciences Centre, London, ON N6A 5W9, Canada; 2Department of Pathology and Laboratory Medicine, Western University, London, ON N6A 3K7, Canada

**Keywords:** epilepsy, review, NGS, epigenomics, DNA methylation

## Abstract

Epilepsy is a highly prevalent neurological disorder, affecting between 5–8 per 1000 individuals and is associated with a lifetime risk of up to 3%. In addition to high incidence, epilepsy is a highly heterogeneous disorder, with variation including, but not limited to the following: severity, age of onset, type of seizure, developmental delay, drug responsiveness, and other comorbidities. Variable phenotypes are reflected in a range of etiologies including genetic, infectious, metabolic, immune, acquired/structural (resulting from, for example, a severe head injury or stroke), or idiopathic. This review will focus specifically on epilepsies with a genetic cause, genetic testing, and biomarkers in epilepsy.

## 1. Genetic Epilepsies: An Introduction

Approximately 30% of all epilepsies are estimated to have genetic etiology [1]. According to the International League Against Epilepsy’s (ILAE) 2017 classification of epilepsies, genetic epilepsies are defined as epilepsies that result directly from a known or presumed known genetic condition and in which seizures are a core symptom of the disorder [2]. To expand on this definition, an epilepsy may be classified as having a genetic etiology in one of three ways: (a) based solely on family history, (b) using clinical research, or (c) if a genetic variant (including DNA sequence or larger chromosomal structural variation) is reproducibly associated with an epilepsy phenotype (Figure 1) [2,3]. In 2022, the ILAE further updated the epilepsy classification system to separately describe epilepsy syndromes based on age of onset [3,4,5,6]. While this is a very important distinction for clinicians, for the purpose of this review all genetic epilepsies will be considered together, regardless of their age of onset.

According to ILAE, genetic epilepsies include both those with monogenic (with either inherited or de novo, meaning not inherited, pathogenic variants) or complex (polygenic with or without environmental factors) inheritance [2]. It is important to note that the updated ILAE classification of genetic epilepsies no longer requires evidence of genetic inheritance, due to a major involvement of de novo pathogenic variants, especially in the context of severe and early onset presentations of epilepsy [2]. For example, Dravet syndrome is a severe developmental and epileptic encephalopathy (DEE) with approximately 80% of cases being caused by mutations in *SCN1A*, many of which are de novo [7]. It has been reported that in infants with severe DEE (including Dravet), de novo variants are causative in up to 30–50% of cases [8]. In a retrospective study of nearly 2000 patients with neurodevelopmental disorders and with epilepsy, 33 genes were observed to have significant enrichment of de novo variants, three of which had limited or no previous evidence of disease association: *CACNA1E*, *SNAP25*, and *GABRB2* [9].

The first genes identified as having an association with epilepsy etiology, including both de novo and inherited etiology, were identified in genes that encode various subunits of neuronal ion channels [10]. Mutations in these ion channel genes allow for the recurrence of seizures due to either a reduction of inhibitory mechanisms or neuronal hyperexcitability [11,12]. Ion channel genes can be broadly divided into two classes: (a) voltage-gated or (b) ligand-gated. Voltage-gated ion channels typically allow specific ions to pass through the cell membrane in response to changes in electrical changes in the membrane potential near the channel [11]. Examples of some voltage-gated ion channel genes associated with epilepsy include: *SCN1A* (encodes a voltage-gated Na+ channel subunit, NaV1.1), *SCN2A* (encodes a voltage-gated Na+ channel subunit, NaV1.2), *KCNQ2* (voltage-gated K+ channel subunit, KV1.2), and *CACNA1A* (voltage-gated Ca2+ channel subunit, CaV2.1) [10]. Like voltage-gated channels, ligand-gated channels allow ions to pass through the membrane, but their response is mediated by the binding of chemical messengers [13]. *GABAA* receptors are members of the cys-loop family of ligand-gated ion channels and are one of the primary mediators of rapid inhibitory synaptic transmission in the central nervous system [14]. Unsurprisingly, variants in several GABA_A_ receptor subunit genes, including *GABRA1*, *GABRB2*, *GABRB3*, *GABRD*, and *GABRG2*, have been associated with several different genetic epilepsies [14,15].

Expanded access to genetic testing for epilepsy, thanks in part to the accessibility of next-generation sequencing (NGS) and more recently whole-exon sequencing (WES) and whole-genome sequencing (WGS), has enabled the identification of many genes encoding proteins other than ion channels that are associated with monogenic epilepsy. See Appendix A for a list of the Ontario Epilepsy Genetic Testing Program’s targeted epilepsy NGS panel. Note the variability of the protein’s function encoded by the different genes.

Amongst the most well-known examples of non-ion channel-associated epilepsy genes are *TSC1* and *TSC2*, components of the mammalian target of rapamycin (mTOR) pathway [16]. The mTOR pathway is a complex and extensively researched biochemical signaling pathway related to cellular proliferation [17]. Pathogenic variants in any number of genes involved in the mTOR pathway have been linked to focal epilepsy, both with and without brain malformations. Mutations in either *TSC1* or *TSC2* contribute to the multisystem disorder tuberous sclerosis [17,18]. Phenotypically, tuberous sclerosis has a wide clinical spectrum, with some patients presenting with minimal symptoms and no neurologic disability [18]. However, more than 80% of patients with tuberous sclerosis have seizures and other symptoms such as multiple cortical dysplasia [19]. Tuberous sclerosis is one example of monogenic epilepsy, where a single gene is considered to be the cause of epilepsy. *TSC1* and *TSC2* display an autosomal dominant pattern of inheritance, which means that in a tuberous sclerosis patient, a single pathogenic variant can be considered causative [18,19]. There are many examples of monogenic epilepsies involving genes with autosomal recessive patterns of inheritance, in which case two pathogenic variants in trans (on two separate alleles or gene copies) are required to cause the epilepsy phenotype.

There are also many examples of genes with an X-linked inheritance pattern that contribute to epilepsy. *NEXMIF* (previously called *KIAA2022*) is inherited in an X-linked dominant (XLD) fashion and like other epilepsy genes is associated with a phenotypic spectrum and has been reported in both male and female patients [20]. It has been reported that 99% of patients with pathogenic *NEXMIF* variants have developmental/intellectual delay and 83% experience seizures [20]. Another X-linked gene, *ARX,* is inherited in an X-linked recessive (XLR) manner [21]. The most frequent pathogenic mutations in *ARX* occur in polyA tracts in exon 2 (the first is a duplication of 24 base pairs and the second is the expansion of a trinucleotide repeat) and while they also can present with a variable phenotype, they have been implicated in Partington syndrome, syndromic and nonspecific X-linked mental retardation, West Syndrome, and intellectual delay [21,22].

### 1.1. Phenotypic and Genetic Heterogeneity in Epilepsy

One of the challenges in identifying a single genetic cause in patients with epilepsy is due to the phenotypic heterogeneity of most of these genes, as well as the genetic heterogeneity in many of the syndromes. Genetic heterogeneity occurs when a similar clinical presentation or phenotype is underpinned by different genetic mechanisms [23]. One subtype of DEE, infantile spasms (IS), is a genetically heterogeneous epilepsy exhibiting a wide range of clinical phenotypes arising from pathogenic variants in a number of genes including *GNAO1*, *KCNB1*, *KIF1A*, *SLC35A2*, *STXBP1*, *TBL1XR1*, and others [24]. Another example is Dravet syndrome which is predominantly caused by variants in *SCN1A*, but other genes have also been associated with the syndrome, such as *GABRA1*, *STXBP1*, *PCDH19* (females only), and *GABRG2* [25,26,27,28].

In contrast to genetic heterogeneity, phenotypic heterogeneity occurs when variants in the same gene result in varying clinical presentations in different patients. NGS studies have revealed how mutations in the same gene can give rise to a spectrum of epilepsy phenotypes (as well as other forms of neurodevelopmental disease) [17]. This can be due to different mutations causing different effects on the translated protein or downstream pathway, leading to either a gain or loss of the protein’s function (Figure 2B) [29]. Understanding the functional implications a pathogenic variant has on the protein or other effects on the downstream pathway is critical for the selection of the correct targeted therapeutic strategies, as well as for the discovery of novel therapies. 

A study investigated variants in *KCNA1* by comparing the functional effects of variants in the pore and voltage sensor region of the encoded Kv1.1 protein [30]. They found that of the four variants studied, three in the pore region had a loss of function (LoF) effect, whereas the variant in the voltage sensor region displayed a hyperpolarizing shift of the activation process, suggesting a gain of function (GoF) affect [30]. They noted that the GoF variant had a milder phenotype and an excellent clinical response to the drug carbamazepine compared to the LoF variants [30].

Another intriguing example of the phenotypic heterogeneity can be seen in the overlap of Dravet syndrome and generalized epilepsy with febrile seizures plus (GEFS+). In addition to being the most common gene associated with Dravet syndrome, *SCN1A* is also the most common causative gene in GEFS+, despite GEFS+ having a significantly less severe clinical presentation [31]. There is no clear genotype–phenotype correlation to explain this phenomenon, although truncating (often presumed to be LoF mutations) and missense mutations in the pore region of the protein tend to be more common in the more severe Dravet phenotype [7]. It is also possible for a GEFS+ phenotype to be caused by a de novo *SCN1A* mutation in one generation, only for the proband’s progeny to inherit the same variant and have a much more severe phenotype, up to and including Dravet syndrome [31,32]. Some of the proposed mechanisms for this observed phenotypic variability are focused on reduced penetrance or are based on the idea of more common risk variants in modifier genes (multiple susceptibility variants) [31,33]. Currently, the theories remain unproven.

### 1.2. Monogenic and Complex Epilepsies

In contrast to phenotypically heterogeneous epilepsies, some epilepsies present with very specific and characteristic clinical and diagnostic imaging findings [34]. These include clinically recognizable syndromes such as Rett and Rett-like syndromes (*MECP2*), Koolen-de Vries (*KANSL1*), or Wolf–Hirschhorn (WHS, caused by copy-number variants on chromosome 4p16.3) [35,36,37]. Collectively, monogenic epilepsies make up a small percentage of genetic epilepsies; however, they are typically associated with some of the earlier onset and more severe clinical presentations of the disease. Some reports have indicated that a monogenic etiology can be identified in up to 40% of patients with severe epilepsy [38].

Apart from rare monogenic forms of epilepsies, the majority of genetic epilepsies are complex genetic epilepsies, where a presumed combination of multiple susceptibility genetic variants is believed to underlie the disease [39]. This is an emerging area of research that is not as well described as other aspects of epilepsy genetics, but with increasing access to WES and WGS, complex genetic epilepsies appear to be an optimistic diagnostic avenue. Recent studies have provided evidence that complex genetic epilepsies may account for a larger percentage of epilepsies than is currently reported. For example, even in acquired epilepsies (such as from trauma or a stroke) there is growing evidence (primarily considering family history) that supports the role that genetic factors play in disease progression despite no specific variant or variants being identified [40,41]. Similarly, the risk to first-degree relatives of a proband with epilepsy is 5- to 10-fold greater than the general population to develop epilepsy following a traumatic brain injury [41].

Another important group of complex genetic epilepsies is genetic generalized epilepsies (GGEs), which account for 15–20% of all epilepsies [42]. GGEs were previously described as idiopathic generalized epilepsies, but the term idiopathic was replaced by genetic in the new ILAE classification of the epilepsies, due to the recognition of genetic variants’ contribution to this group of epilepsy [2,39]. Adding to the complexity, a patient may present initially with one GGE syndrome (for example, childhood absence epilepsy), only to later evolve into a second subtype (such as juvenile myoclonic epilepsy) [43]. It has been suggested that the underlying pathophysiology of GGE may be related to the corticothalamic relay system, as many of the implicated genes are involved in this pathway [33]. For example, the T-type calcium channel is expressed in the thalamus, and variants in genes encoding T-type calcium channel subunits, such as *CACNA1H*, have been observed in GGE patients [44]. While the underlying cause of GGEs remains unclear, a polygenic inheritance pattern seems likely given some observations: most probands report no Mendelian pattern of inheritance, yet their first-degree relatives have been reported to be 8% more likely than the general population to develop epilepsy [45]. Though elevated, the risk is still much lower than would be expected with Mendelian segregation (25% for an autosomal recessive inheritance pattern, 50% for an autosomal dominant) [45]. Polygenic or multifactorial inheritance is often associated with the more common, less severe, and later age onset epilepsies [1,33]. Molecular testing of low penetrance susceptibility genes, at this time, provides limited clinical diagnostic utility and predictive value for the onset and severity of epilepsy-related symptoms [33].

Another piece of evidence for a polygenic inheritance pattern is the observation that a combination of common variants (i.e., the same variants reported in multiple, unrelated families) have been reported to have a role in GGE etiology [46]. This is in contrast to monogenic focal epilepsies, which appear to be caused predominately by rare variants (i.e., unique variants reported in a limited number of families) [46]. However, a monogenic cause can be identified in a small percentage of patients with GGE and a significant family history. For example, in a pedigree remarkable for childhood absence epilepsy and febrile seizures (a subset of GGE), a missense variant in *GABRG2*, encoding the γ-aminobutyric acid (GABA_A_) receptor γ2 subunit was identified and described as causative [47]. Thus, genetic testing may benefit some families with GGE who present with a strong family history with suspected Mendelian inheritance. Another example of complex genetic epilepsy and the action of modifiers can be observed by the growing evidence that some large, recurrent chromosomal deletions/duplications may be potential epilepsy susceptibility factors. While there is not enough evidence to support these as being directly causative, 15q13.3, 15q11.2, and 16p13.11 copy-number variants (CNVs) have been detected in patients with focal, GGE, or epileptic encephalopathy more frequently compared to the general population [12]. The 15q13.3 microdeletion has been associated with a 34-fold risk for developing GGE (Figure 2A) [48]. While 15q13.3del CNVs have also been associated with childhood-onset absence seizures and can also present with atypical symptoms such as resistance to anti-seizure medications (ASMs) or intellectual disability, only about one third of people with this CNV have epilepsy [49]. 16p11.2 deletions and duplications affect approximately 3 in every 10,000 individuals and epilepsy or seizures have been reported in approximately 25% of patients with a deletion and in up to almost 30% of duplication carriers [50]. The critical region for the 16p11.2del spans 500–600kb and contains 27–29 genes [50,51]. While rare, collectively, CNVs (including microdeletions and microduplications) are estimated to contribute to at least 10% of childhood-onset epilepsies and up to 5% of all epileptic encephalopathies [52]. In addition to microdeletion/duplications, other large chromosome alterations have been described in association with epilepsy. For example, ring chromosome 20 is a well-documented chromosomal disorder associated with focal motor and dyscognitive seizures [53].

## 2. Genetic Testing in Epilepsy

Advancements in genomic testing technologies have contributed to the identification of a broad range of genetic causes of epilepsy including monogenic alterations, polygenic/risk factor variants, as well as large chromosome alterations including CNVs. Chromosome microarrays (CMA) are often used to test for microdeletion/microduplication syndromes associated with epilepsy phenotypes. These large CNVs typically involve deletions or duplications of several genes and are associated with a wide spectrum of phenotypes, even within a specific region. For example, the typically recognizable epilepsy syndrome Wolf–Hirschhorn syndrome (WHS) can be defined into three phenotype classes generally correlated with the size of the deletion [37]. Firstly, a likely underdiagnosed mild phenotype is associated with the smallest deletion, typically not more than 3.5 Mb in size [37]. The second and most common subtype is associated with the core WHS phenotype and has been reported in patients who carry larger deletions between 5–18 Mb in size [37]. Larger CNVs in this region, typically >22–25 Mb, cause a much more severe phenotype [37]. CNVs in any of the WHS subtypes include not only simple deletions (often de novo), but complex rearrangements (predominantly unbalanced translocations) are also possible etiologies [37,54]. Interestingly, the core phenotype of WHS has been reported to have a critical region as small as 1.9 Mb on the terminal of 4p [54]. Presentation of either the mild, core, or severe phenotype of WHS is therefore not only dependent on the size of the CNV, but also of the gene content that is affected by the CNV.

Often subsequent to microarray analysis, targeted NGS panels are also used for the diagnosis of genetic epilepsies predominantly with monogenic causes [33,55]. NGS panels can target hundreds of genes associated with monogenic epilepsy. Panels can be designed to detect both single nucleotide variants in gene coding regions and small copy-number variants (exon/gene level) within a single assay. For example, the Ontario Ministry of Health Epilepsy Working Group recommended and implemented the Ontario Epilepsy Genetic Testing Program (OEGTP), a 167 gene panel for testing of Ontario patients commencing in October 2020 [56]. The current molecular diagnostic yield is approximately 20% in this population, demonstrating the valuable clinical utility of genetic testing in the diagnosis of epilepsy. See Appendix A for a breakdown of included subpanels and the full complement of genes included.

### 2.1. Importance and Limitations of Genetic Testing in Epilepsy

While significantly expanding the diagnostic yield, NGS panels also have some limitations. NGS targeted panels are not designed to identify causative variants located in deep promoter or intronic regions (the clinical standard based on the ACMG guidelines) and are typically designed to cover exons and +/−20 basepairs of the flanking introns, and only sometimes select promoters [57]. Targeted NGS panels are also unable to detect variable number tandem repeats (VNTRs), which can be disease causing in a number of monogenetic epilepsies. This is true for repeats located outside of the region targeted by the panel, such as in familial adult myoclonic epilepsy 1 (FAME1) where intronic repeat expansion variants in *SAMD12* are associated with epilepsy [58]. Furthermore, standard NGS (short read) technology is also unable to detect repeat expansions, even if they are located within the NGS panel design such as ones causing Unverricht–Lundborg type epilepsy (also known as epilepsy, progressive myoclonic 1A) where repeat expansions are located in a promoter of the *CSTB* [59]. While Unverricht–Lundborg type epilepsy is a relatively rare disorder, it is estimated that 99% of variants in *CSTB* are due to this repeat expansion [59,60]. Despite these limitations, targeted NGS panel testing has been reported to provide a diagnostic rate of 15–25% (variability is largely due to family history, clinical presentation, and age of onset) [61,62]. Another challenge involves a limited understanding of growing numbers of newly identified gene sequence variations and our ability to interpret their clinical significance, which is exceedingly evident in large-scale whole-genome and exome sequencing (WES and WGS) approaches. This is a barrier to epilepsy diagnosis and management, as the correct and timely identification of genetic epilepsies is a critical piece of patient care and can have a direct impact on patient outcomes.

Genetic mosaicism presents another technical diagnostic challenge. There is a strong association between the level of mosaicism in a patient’s tissues and the severity of their phenotype [63,64]. Recent studies have indicated that somatic variants confined to the brain may contribute to the yet largely undescribed genetics of focal epilepsies, particularly in genes associated with the mTOR pathway [64]. There have been some reports of the presence of both germline and somatic mutations in repressors of the mTOR pathway, in line with a biallelic loss or the so called “two-hit” genetic model [65]. Somatic mutations causing brain disease can sometimes be found in low abundance outside the brain, detected in either buccal cells, blood-derived DNA, or in cell-free DNA from plasma; these, may prove to be useful targets as biomarkers in the future [46,64].

There is emerging evidence that it is cost effective to perform these and other genomic tests earlier, as it may reduce the overall costs associated with repeat imaging or EEGs as well as more invasive testing, such as muscle biopsies or repeat specialist visits [33,55]. In addition to this diagnostic utility, genetic testing can provide also prognostic and clinical management information. For example, epilepsy surgeries have less favorable outcomes in patients with pathogenic variants in ion channel genes such as *SCN1A* [66].

### 2.2. Therapeutic Strategies in Monogenic Epilepsies

With the discovery of many genes involved in epilepsy, precision therapies for monogenic epilepsies have rapidly developed in recent years. These therapies are designed to target the patient’s specific genetic alteration and include diverse approaches, such as anti-sense oligonucleotides (ASO), RNA interference (RNAi), inhibitors of cellular signaling, and modulators of metabolism [67].

In certain cases, mutations in enzyme-encoding genes can be targeted by metabolic modulation through dietary intervention (Figure 2B). One example involves *SLC2A1* mutations resulting in GLUT1 deficiency. GLUT1 is a key transporter required for glucose penetration across the blood–brain barrier and its deficiency results in a lack of glucose in the brain for energy production and consequently neuronal dysfunction [68,69]. Patients with GLUT1 deficiency respond well to the ketogenic diet, most likely because ketones provide an alternative source of energy for the brain, bypassing the metabolic defect and ultimately leading to improved seizure control, and in some cases, intellectual abilities [68,69,70].

In epilepsies caused by alterations in mTOR pathway genes, such as *DEPDC5*, *TSC1*, or *TSC2*, the drug rapamycin can be used to manage symptoms as it inhibits mTOR pathway [65,71,72].

Ion channel alterations have been targeted using ion channel modulators, such as sodium channel blockers, as well as gene-silencing ASO approach, for example in *KCNT1*-associated epilepsies [73]. Understanding genetic etiology is essential for the treatment of channelopathies. Patients with *PRRT2*-associated paroxysmal movement disorders respond well to sodium-blocking drugs such as carbamazepine [74]. However, the symptoms of Dravet syndrome can be worsened if treated with sodium-blocking drugs and should therefore be avoided in *SCN1A*-positive patients [75].

While necessary, ASMs are not without side effects and other risks, and must be selected while considering the seizure and epilepsy types, the epilepsy syndrome (genetic cause), and other adverse effects associated with the drug [76]. For example, sodium valproate (also known as valproate or VPA) is a very commonly prescribed ASM and is also a teratogen [77]. A number of factors such as dosage, the number of co-administered ASMs, hereditary susceptibility, and gestational age of the fetus at exposure contribute to the teratogenicity of VPA [77]. The sodium channel modulators, oxcarbazepine and lamotrigine, are often considered first-line therapy for focal epilepsies [76]. The selection of ASMs in generalized epilepsy can be more challenging and is based on the type of epilepsy syndrome, patient’s demographics (sex, age, and psychiatric history). It is estimated that seizure freedom is achieved in 60–70% of patients [76]. The genotype of epilepsy patients can aid clinicians in balancing the risks and benefits of ASMs in order to best manage their seizures. As the underlying molecular mechanisms of encephalopathies and epilepsies are better classified and understood, further opportunities for the development of novel targeted therapies will improve personalized therapies for these patients [10,33,38,78].

### 2.3. Epilepsy Biomarkers

Given the phenotypic and genotypic heterogeneity of genetic epilepsies, and the clinical utility of definite molecular diagnosis, there is a growing need for effective non-invasive biomarkers for diagnostic screening and reclassification of ambiguous genetic test results. See Table 1 for a summary of reported biomarkers in epilepsy.

One promising diagnostic biomarker is extracellular high-mobility group box 1 (HMGB1) protein. HMGB1 is a proinflammatory mediator that is involved in various neurological disorders [79]. Wang et al. measured HMGB1 concentrations in paired serum and cerebrospinal fluid in patients with drug-refractory epilepsy, newly diagnosed epilepsy, and other non-inflammatory neurological disorders. They and others found no correlations in HMGB1 levels between cerebrospinal fluid (CSF) and serum [79,80]. However, this study reported that the CSF HMGB1 concentrations were significantly higher in the drug-refractory epilepsy and newly diagnosed epilepsy groups compared with the other non-inflammatory neurological disorders group [80]. Also, patients with symptomatic etiology showed significantly higher levels of CSF HMGB1 and expressed elevated levels of CSF HMGB1 at one-year follow up, only in patients who did not go into remission [80]. Additionally, the group found that the CSF HMGB1 levels were positively associated with seizure frequency [80,81].

**Table 1 ijms-24-14606-t001:** Emerging biomarkers and their potential utility reported in epilepsy.

Biomarker	Sample Type	Potential Utility	Examples	Current Limitations
High-mobility group box 1 protein	Primarily cerebrospinal fluid (CSF), some serum	CSF concentrations were significantly higher in drug-refractory epilepsy and newly diagnosed epilepsy groups compared with other non-inflammatory neurological disorders groups [79]. Levels in CSF reported to be positively associated with seizure frequency [81].	HMGB1	Conflicting reports for the utility of HMGB1 in blood, and the more invasive CSF remains preferred sample type
Neurofilaments	Serum or plasma	Elevated levels have been reported in patients with autoimmune epilepsy and in adults with post-stroke epilepsy when compared to single-seizure patients [82]. Study involving patients with Down syndrome reported near-significant elevation of NfL levels in patients with epilepsy compared to no epilepsy [83].	NfL	Not specific to epilepsy as NfLs are released into the blood stream following neuronal damage
Purines	peripheral blood	Acute seizures and epilepsy have been reported to be associated with increased blood purine levels [84]. Deficiency of adenosine has been associated with an increase in DNA methylation levels (proposed to be implicated in epileptogenesis) [85].	Adenosine	Not specific to seizures or epilepsy and the short half-life of purines in blood makes it a challenging biomarker target
microRNAs (miRNAs)	plasma	miRNAs reported to be dysregulated in the plasma of patients with intractable temporal lobe epilepsy [86].	miR-93-5p, miR-199a-3p and miR-574-3p	Emerging field
DNA Methylation (Targeted)	brain tissue, peripheral blood	Significant promoter hypermethylation was detected in epileptic patients of some gene promoters when compared between healthy controls [87]. Differentially methylated regions (DMRs) in at imprinting sites has been associated with epilepsy [88].	*CPA6* *UBE3A*	Most targeted screens require knowing the target, which is useful for reclassification of variants, but not for screening
Global DNA Methylation	brain tissue, peripheral blood	Altered DNA methylation patterns in mesial temporal lobe epilepsy detected when compared to unaffected controls [89]. Blood DNA methylation episignatures have been described for syndromic syndromes with epilepsy [90].	*ATRX*, *CHD2*, *EHMT1*, *KANSL1*, *KDM5C*, WHS CNVs	Emerging field and limited episignatures exist for non-syndromic epilepsies

Neurofilaments (NfL) are axonal proteins that maintain the structure of neurons and are released into the CSF and bloodstream following neuronal damage, and they appear to be another promising biomarker candidate for epilepsy [82,91]. NfL is an established clinical marker of a number of neurodegenerative diseases, such as multiple sclerosis, Parkinson’s disease, and amyotrophic lateral sclerosis [82]. While the work that discusses NfL’s utility as a serum or plasma biomarker in epilepsy is limited, elevated serum levels have been reported in patients with autoimmune epilepsy [92]. A study involving patients with Down syndrome also reported near-significant elevation of NfL levels in patients with epilepsy compared to controls [83]. Elevated serum concentrations were also detected in adults with post-stroke epilepsy when compared to single seizure patients, but not in patients with non-stroke related epilepsy [93].

Elevations in blood purine concentrations following experimental seizures and during epilepsy in humans have also been reported. Blood purine levels correlated with both seizure severity and brain damage and were able to distinguish patients with epilepsy from unaffected controls [84]. However, the short half-life of purines in blood has proved technically difficult and labor intensive to assay in the past. Using the “SMARTChip”, Beamer et al. reported that acute seizures and epilepsy are associated with increased blood purine levels [84]. The authors noted that a major limitation of their study is that increased blood purine levels are not specific to seizures or epilepsy and have been previously reported in post-brain injury, post-ischemia, and post-hypoxia cases [84]. Despite this, enzymatic detection of blood purines has a good potential for supporting early clinical decisions regarding the monitoring and treatment of seizures and epilepsy.

One purine in particular, adenosine, has also been identified as a potential therapeutic target, via the ketogenic diet. There is evidence that ketones (a byproduct of the modified diet) not only provide an alternative source of energy for the brain, but that adenosine also plays a critical role as an endogenous regulator of DNA methyltransferases [85]. Adenosine is the obligatory end product of S-adenosylmethionine (SAM) dependent methylation reactions and either the upregulation of adenosine kinase expression or a deficiency of adenosine has been associated with an increase in DNA methylation levels, which has been proposed to be implicated in epileptogenesis [85,94]. There has been a positive correlation between the ketogenic diet and outcomes for epilepsy patients, especially in drug-refractory epilepsies, which encompass an estimated 30% of epilepsies, with one study reporting more than 90% seizure reduction among those on the diet [94,95].

Many studies have demonstrated the role of another epigenetic mechanism, microRNAs, in the pathogenesis of epilepsy [96]. MicroRNAs are short non-coding RNA molecules that negatively regulate gene expression [97]. The relative ease of detection of microRNAs in biofluids, resistance to degradation, and evidence supporting their functional role in epilepsy make them excellent candidate biomarkers [97,98]. A multi-model, genome-wide profiling of plasma microRNAs during epileptogenesis and in chronic temporal lobe epilepsy animal models identified a number of biomarker candidate microRNAs including increased miR-93-5p, miR-142-5p, miR-182-5p, and miR-199a-3p expression and decreased miR-574-3p [86]. Further validation studies found miR-93-5p, miR-199a-3p, and miR-574-3p were also dysregulated in the plasma of humans with intractable temporal lobe epilepsy [86]. Interestingly, treatment of affected mice with common ASMs did not alter the expression levels of any of the five microRNAs identified; however, administration of an anti-epileptogenic microRNA treatment maintained baseline expression of several of these microRNAs [86]. The microRNAs were detected within the Argonuate2-RISC complex from both neurons and microglia, which the authors suggested could mean these miRNA biomarker candidates could be traced back to specific brain cell types [86,97]. Studies have identified additional circulating microRNA biomarkers of both experimental and human epilepsy, which may eventually support the diagnosis of temporal lobe epilepsy via a quick, cost-effective, and rapid molecular-based test.

Another epigenetic mechanism that is emerging as a promising area of biomarker research involves changes in DNA methylation. DNA methylation refers to the addition of a methyl group to cytosine bases by a methyltransferase enzyme [99]. Aberrant DNA methylation has been associated with a number of diseases, including epilepsy [100]. One report found that both the methylation status of the UPS29 minisatellite of the *ACAP3* gene and the length of a repeat polymorphism of the same region could serve as a biomarker in epilepsy [101]. *ACAP3* is predicted to be involved in neuron migration and the regulation of neuron projection development [102]. Most notably, Suchkova et al. described that the frequency of individuals with hypomethylated UPS29 alleles increased statistically significantly in both men and women in comparison with controls among patients with symptomatic epilepsy [101]. They also found that the short UPS29 allele and hypomethylation of this tandem repeat may be new genetic/epigenetic markers for some epilepsies, though this finding was not consistent between men and women.

Another study investigated the carboxypeptidase A6 (*CPA6*) gene, associated with a recessive familial form of febrile seizures and temporal lobe epilepsy and in sporadic temporal lobe epilepsy [87]. When the DNA methylation status of the *CPA6* promoter was compared between healthy controls and patients affected by epilepsy, a significant promoter hypermethylation was detected in epileptic patients [87].

Aside from gene-specific changes in DNA methylation, altered global DNA methylation has also been reported in epilepsy. Differences in DNA methylation profiles have been reported between brain tissue samples collected from patients with temporal lobe epilepsy–hippocampal sclerosis ILAE Type I compared to temporal lobe epilepsy with non-hippocampal sclerosis using whole-genome bisulfite sequencing [103]. The study identified potential cellular signaling pathways associated with differentially methylated regions at the whole-genome level, as well as in the context of promoter regions [103]. Although epilepsy surgery is relatively common, especially in temporal lobe epilepsy, it is a relatively invasive procedure that limits the utility of brain-derived biomarkers.

Epilepsy-specific DNA methylation profiles have also been reported in blood. Using Illumina 450K methylation microarrays, blood DNA methylation patterns have been observed as being altered in mesial temporal lobe epilepsy when compared to unaffected controls [89]. While mesial temporal lobe epilepsy has not been historically classified as a genetic epilepsy, regions of hyper- and hypomethylation were identified in both promoter and coding regions of genes that were associated with pathways predicted to participate in growth regulation (such as skeletal development), ion binding, oxidoreductant activity, and drug metabolism, offering evidence that a genetic component may play a yet undefined role in its epileptogenesis [89].

Also using Illumina 450K and EPIC methylation microarrays, it has been demonstrated that patients with an expanding number of different genetic conditions can be differentiated from healthy controls by their global DNA methylation patterns in blood [88,90]. DNA methylation episignatures, defined as a recurring epigenomic pattern associated with a common genetic or environmental etiology in a particular patient population, have also been clinically validated for many genetic epilepsy syndromes including *ATRX*, *CHD2*, *EHMT1*, *KANSL1*, and *KDM5C* [90]. Pathogenic mutations in *KDM5C* cause Claes–Jensen syndrome, an X-linked disorder wherein males with hemizygous mutations are affected and the majority of heterozygous carrier females are either unaffected or more mildly affected than hemizygous males with the same variant [104,105]. The milder phenotype of *KDM5C* females seems to correlate with the milder *KDM5C* episignature seen in females, perhaps hinting at an underlying epigenetic role. Given that *KDM5C* is a lysine-specific demethylase, this would not be surprising but requires further investigation [105]. The *KDM5C* methylation episignature is one unique example of biomarker that can distinguish carrier or mildly affected females and males from unaffected controls [105]. Syndromes caused by CNVs, such as WHS (described above) have also been associated with episignatures [90]. Screening for these and a number of other neurodevelopmental disorders is available as a clinical test, EpiSign™ [90]. EpiSign™ testing is also able to screen known differentially methylated regions (DMRs) in regions subject to imprinting where abnormal DNA methylation has been associated with epilepsy, such as Angelman syndrome (causative gene: *UBE3A*) [88]. These episignatures have been shown to be highly sensitive and specific to each syndrome [88,90]. In genetic epilepsies as with other neurodevelopmental disorders where overlapping clinical phenotypes and a wide range of genomic associations make accurate diagnosis a challenge, the sensitivity and specificity of a biomarker is critical [106]. In addition to being a potential diagnostic tool for epilepsies, episignatures can also offer functional evidence to aid in variant reclassification. This is of particular utility when assessing variants of unknown significance, the prevalence of which are only expected to increase in number with broader utilization of WES and WGS.

Recent studies in epilepsy have demonstrated that the zebrafish is a useful model organism for epilepsy research given the high degree of conservation between human epilepsy associated genes and their zebrafish orthologs [107]. Functional testing of variants of unknown significance or drug screening with this animal model (as well as other in vitro, meaning experiments occurring outside a living organism, and in vivo, experiments inside a living organism, models of epilepsy) have the potential to help diagnose more patients. It also seems very likely that the future direction of epilepsy genetics will include the expanded use of WGS and will need to include techniques involving the other -omics (i.e., epigenomics, transcriptomics, metabolomics) to improve our understanding of epilepsy and to provide better treatment options for those affected with the disease.

In particular, epigenomic changes, such as changes in DNA methylation or microRNA expression may hold the answers to many of the questions faced by epilepsy researchers. As shown here, epilepsy is an incredibly complex disorder, and it stands to reason that environmental factors might play a crucial role in epilepsy’s progression via epigenetic modification. It is plausible that a portion of the phenotypic and genetic heterogeneity seen in epilepsy is caused by yet undefined epigenomic alterations. However, given the complexity associated with epilepsy, it seems unlikely that any one mechanism will emerge as the dominant cause. In fact, a global hypermethylation pattern has also been identified in a number of miRNA and long non-coding RNA (lncRNA) genes in TLE patients, suggesting that multiple levels of epigenomic regulation may operate together [108].

Hence, epigenomic mechanisms are a promising avenue for the discovery of diagnostic biomarkers which may also offer further unique insights into the ethology of epilepsy. These findings could offer biological explanations for epileptogenesis, disease progression, or to why some epilepsies become refractory to ASM, but others respond well to medication. The development of epigenomic biomarkers in non-invasive tissues such as blood is a promising avenue for the molecular diagnosis of genetic and environmental causes of epilepsies. Therapeutic strategies including epigenetic pathways such as adenosine and the SAM-dependent DNA methylation pathway are expanding, and it is likely that further epigenomic studies will only highlight additional diagnostic and prognostic biomarkers as well as treatment modalities involving epigenetic pathways involving genetic and environmentally cased epilepsies.

## Figures and Tables

**Figure 1 ijms-24-14606-f001:**
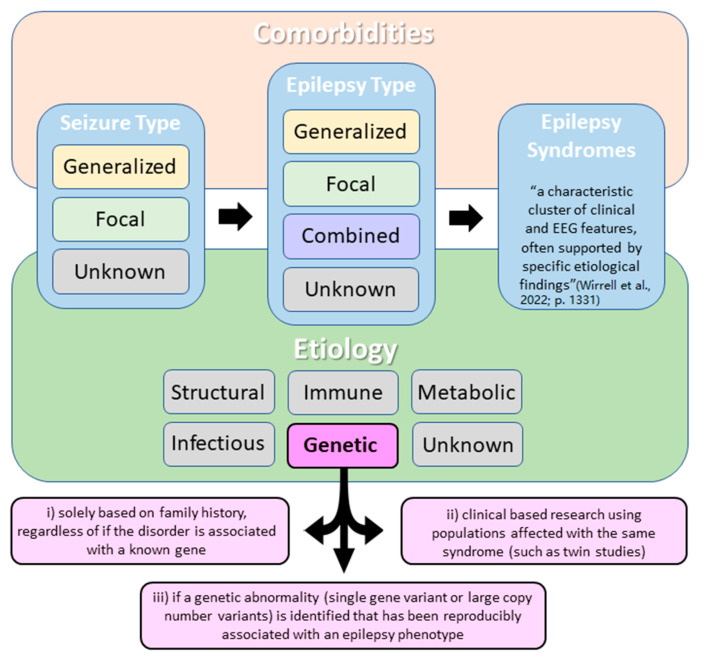
Simplified International League Against Epilepsy (ILAE) classification of the epilepsies with a focus on epilepsies with a genetic etiology. According to the ILAE, three diagnostic levels should be considered when classifying epilepsy: (a) seizure type, (b) epilepsy type, and (c) epilepsy syndrome. Etiology and comorbidities should be considered at each diagnostic level. Note that epilepsy may be classified as having a genetic etiology in one of three ways (as described in pink squares) [3].

**Figure 2 ijms-24-14606-f002:**
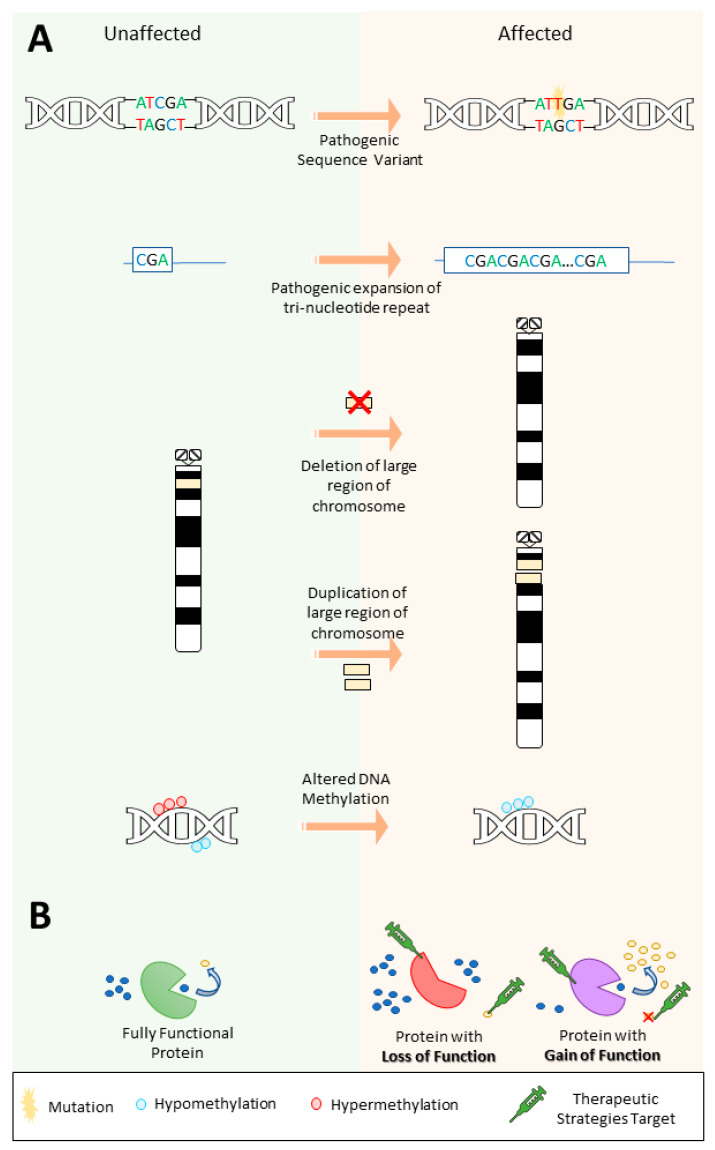
Types of genetic variants that contribute to epilepsy and potential targets for therapeutic strategies depending on the effect of the pathogenic variant. (**A**) Different pathogenic variants can cause epilepsy, including (i) sequence variants, (ii) expansion repeat variants, (iii) copy-number variants, and (iv) epigenetic variation, such as altered DNA methylation. Note that copy-number variants can also include gene or intra-gene level changes. Also, different types of variants can also act together, for instance, a repeat expansion may cause increased hypermethylation of a gene promoter leading to its inactivation. (**B**) The effect a variant has on downstream pathways can lead to either a loss or gain of function and therapeutic strategies can target the affected protein as well as substrates or downstream products in the pathway depending on the effect of the variant.

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
