# Peer review of "Complexity in Genetic Epilepsies: A Comprehensive Review"

_ijms, 2023, doi:10.3390/ijms241914606_

Round 1

Reviewer 1 Report

In this review, the authors summarise genetic causes, genetic testing and biomarkers for epilepsy. On the whole, it is well-written and organized and easy to follow. It provides an in-depth information of the state-of art of genetic epilepsy.

However, some minor changes and the development of some points would improve the quality of this review and would help the readers to better understand some concepts that have been given without accurate details.

1. The authors should mention additional pathological genes with a crucial role in epilepsy. For example SCN2A, an important pathogenic gene whose mutations can cause allelic diseases such as malignant DEE and benign familial DEE; ARX, an X-linked triple trinucleotide repeat gene, has been found in male children with childhood spam; and Nexmif, an X-linked gene found mutated in both male and female patients.

2. The authors should clarify that different mutations in the same gene can cause different disease phenotypes because they modify the activity of the affected protein or pathway differently. Also, illustrative images should be included to understand this point.

3. The authors report on the applicability of VPA as anticonvulsant epidrug. Other epidrugs have been proposed for their anti-epileptic activity. The authors should add a comment.

4. Concerning the X-linked KDM5C gene, the authors fail to clarify that frequently - as testified by numerous studies – the heterozygous carrier females have a less severe disease phenotype than the hemizygous carrier males. The authors should mention recent articles related to this point.

Reviewer 2 Report

Rastini and colleagues provide a review on genetic caused epilepsies including testing and biomarkers.

The manuscript is  thought to be included in a special issue ‘The Role of Specific Alteration in Neurological Disorders: From Molecular Mechanisms to Therapeutic Strategies‘ nevertheless the authors only point therapeutic strategies randomly in the manuscript and at least need to be referenced with current guidelines. In general, the manuscript seems to be unstructured e.g. subheading two is missing. Several pages under one subheading are too long and need substructures to help the reader. Also a table e.g. for the section biomarkers would be welcome. In fact, the supplemental tables are a nice reference collection for current genetic testing and a reduced table in the main text e.g. for the most frequent affected genes would be great.

Besides that, there are some formatting (e.g. italic written Latin words throughout the text), word repetitions or grammar problems in single sentences (e.g. line 16, Figure 1, line 53 etc.).

There are some formatting (e.g. italic written Latin words throughout the text), word repetitions or grammar problems in single sentences (e.g. line 16, Figure 1, line 53 etc.).

Round 2

Reviewer 2 Report

The authors improved the manuscript according to the reviewer’s comments. Nevertheless, there are some minor points to address:

- There are still word unnecessary word repetitions again ‘… with a genetic cause, including causes, genetic…’, ‘…based on clinical based research…’

- When writing gene names italic from the nomenclature it is clear that the gene is meant, therefor to write ‘MECP2 gene’ is unnecessary, write ‘MECP2’, also for other examples in the text.

here are still word unnecessary word repetitions again ‘… with a genetic cause, including causes, genetic…’, ‘…based on clinical based research…’

- When writing gene names italic from the nomenclature it is clear that the gene is meant, therefor to write ‘MECP2 gene’ is unnecessary, write ‘MECP2’, also for other examples in the text.

Author Response

We thank the reviewer for their helpful feedback and have implemented their suggestions and requested changes, which we feel further strengthen the manuscript. Please see the attached document.
